# The Contribution of the 20S Proteasome to Proteostasis

**DOI:** 10.3390/biom9050190

**Published:** 2019-05-16

**Authors:** Fanindra Kumar Deshmukh, Dana Yaffe, Maya A. Olshina, Gili Ben-Nissan, Michal Sharon

**Affiliations:** Department of Biomolecular Sciences, Weizmann Institute of Science, Rehovot 7610001, Israel; fanindra@weizmann.ac.il (F.K.D.); dana.yaffe@weizmann.ac.il (D.Y.); maya.olshina@weizmann.ac.il (M.A.O.); gili.ben-nissan@weizmann.ac.il (G.B.-N.)

**Keywords:** protein degradation, 20S proteasome, cellular homeostasis, oxidative stress, intrinsically disordered proteins, neuronal communication

## Abstract

The last decade has seen accumulating evidence of various proteins being degraded by the core 20S proteasome, without its regulatory particle(s). Here, we will describe recent advances in our knowledge of the functional aspects of the 20S proteasome, exploring several different systems and processes. These include neuronal communication, post-translational processing, oxidative stress, intrinsically disordered protein regulation, and extracellular proteasomes. Taken together, these findings suggest that the 20S proteasome, like the well-studied 26S proteasome, is involved in multiple biological processes. Clarifying our understanding of its workings calls for a transformation in our perception of 20S proteasome-mediated degradation—no longer as a passive and marginal path, but rather as an independent, coordinated biological process. Nevertheless, in spite of impressive progress made thus far, the field still lags far behind the front lines of 26S proteasome research. Therefore, we also touch on the gaps in our knowledge of the 20S proteasome that remain to be bridged in the future.

## 1. Introduction

The 20S proteasome is a conserved degradation machinery that is essential for maintaining cellular homeostasis [1,2]. The architecture of the 20S proteasome particle is highly conserved, forming a 700-kDa cylindrical structure whose proteolytic active sites are encapsulated within its inner chamber [3]. The 20S proteasome is composed of 28 subunits, arranged in four heptameric rings (α_7_β_7_β_7_α_7_). In eukaryotes, the two outer rings consist of seven distinct α-subunits each. Similarly, seven different subunits form the two inner β-rings, three of which are responsible for the proteasome’s proteolytic activities. The β1 subunit displays caspase-like activity, β2 shows trypsin-like activity, and the β5 subunit possesses a chymotrypsin-like function. Together, these catalytic subunits ensure the degradation of numerous protein substrates. Overall, 20S proteasome architecture creates a compartment whose proteolytic active sites are restricted to its interior, so that only proteins entering this chamber are degraded.

Thus far, the 20S proteasome was mainly seen as a component of the 26S proteasome complex, where it is known to be involved in the degradation of ubiquitin-modified proteins. During this process, the 20S proteasome associates with one or two 19S regulatory particles to form the 26S proteasome cooperative machinery. The 19S particle contributes to substrate recognition, unfolding, and delivery of the unfolded substrate into the 20S proteasome, where proteolysis occurs [3,4]. In addition to this indispensable degradation route, heavily studied over the years, emerging new evidence indicates that degradation can also be mediated solely by the 20S proteasome, without the regulatory task of the 19S particle, or the substrate selectivity achieved by ubiquitin tagging [5]. The functions of this free, uncapped 20S proteasome are the focus of this review.

Biochemical analysis of HeLa cell extracts have revealed that free 20S proteasomes greatly outnumber capped proteasome species, such as the 26S proteasome [6]. In support of these initial findings, recent studies using quantitative proteomics across many different cell types determined that nearly two-thirds of the proteasomes available within the cell exist, are in fact, in the free 20S form [7,8]. Moreover, it has been shown that at least 20% of cellular proteins undergo 20S proteasomal cleavage [9] and, under oxidizing conditions, the 20S proteasome was identified as the major degradation machinery [10,11].

Structure, or more precisely lack of structure, is the criterion for degradation by the 20S proteasome [5]. The 20S proteasome cleaves proteins that contain partially unfolded regions; thus, capable of entering into its catalytic chamber. Two main groups of substrates that fall into this category were shown to be susceptible to 20S proteolysis. The first consists of proteins that have lost their native structure due to aging, mutations, or oxidative damage. These proteins are prone to aggregation and may result in cytotoxicity. Therefore, they should be rapidly removed to prevent human pathologies such as cardiovascular disease, ischemic stroke, and neurodegenerative disorders [12]. The second group comprises substrates whose unfolded regions are an intrinsic feature of the proteins themselves [5]; i.e., proteins featuring intrinsically disordered regions (IDR). However, degradation by the 20S proteasome is not mutually exclusive, and different pools of the same protein can be sent to degradation via either the 20S or the 26S proteasome.

Here, we will discuss the emerging view of how degradation by the 20S proteasome influences cellular processes (Figure 1). In doing so, we will describe the role of the 20S proteasome in neuronal stimulation, antigenic peptide production, and regulation of protein function through post-translational processing. We will also explore the current understanding of 20S-mediated proteolysis of IDR proteins, and the 20S proteasome role in oxidative stress adaptation. Finally, we will review some of the recent findings that underscore the role of the 20S proteasome in the extracellular milieu, on those circulating freely in the plasma, and those encapsulated in extracellular vesicles.

## 2. Adaptation to Oxidative Stress

Cells utilize a broad array of mechanisms to cope with oxidative stress, including drastic reductions in endogenous levels of ATP [13,14,15], which inversely affect the levels of NADPH, a major component in the cellular reactive oxygen species (ROS) detoxification system [16]. Depletion of ATP also inevitably inhibits all ATP-dependent processes, keeping the cells in a semi-dormant state while they cope with damage repair [17]. Moreover, oxidative stress hampers new protein synthesis, the functionality of ATP-independent chaperones, and the synthesis of polyphosphates, which act as inorganic chaperones [17,18,19]. When damage caused by oxidative stress is beyond repair, breakdown of the impaired proteins by means of proteasomal and lysosomal degradation pathways, serves as a last resort [20,21,22].

Oxidized proteins can theoretically be degraded either by ubiquitin/ATP-dependent or -independent mechanisms, through the 26S or 20S proteasome, respectively. The 20S proteasome, however, appears to be the major machinery involved in this process. Initially, redox modifications induce a rapid increase in the catalytic activity and proteolytic capacity of pre-existing 20S proteasomes [23,24,25]. In addition, short-term exposure to oxidative stress induces the disassembly of the 26S proteasome complex into 20S and 19S particles [10], a process that is assisted by the yeast proteasome-interacting protein Ecm29 [26] and its mammalian orthologue, KIAA0368 [27]. Ecm29 knockdown results in a significant reduction in 26S proteasome disassembly upon exposure to oxidative stress, which is accompanied with reduced cell survival rates [26]. On the other hand, mouse KIAA0368 knock-out cells were shown to be more resistant to hydrogen peroxide exposure, in spite of the reduced dissociation of the 26S proteasome into active 20S complexes [27]. Hence, further research is needed, in order to specifically indicate whether the 26S proteasome takes part in eliminating oxidized proteins and whether Ecm29/KIAA0368 is involved in increasing 20S levels by inducing 26S disassembly.

Heat shock protein 70 (HSP70), a critical component of protein quality control in the cell, was also shown to mediate dissociation and then reassociation of the 26S proteasome during cellular adaptation to oxidative stress, by binding and preserving the dissociated 19S regulators for subsequent reassembly into 26S proteasomes [24]. As a consequence, the proteolytic activity of the 26S proteasome is reduced, coupled with an increase in the activity of the 20S proteasome core particle [28,29,30].

The shift in activity levels of the different proteasome complexes makes sense within the context of oxidative stress, since oxidized proteins are, by nature, partially unfolded and, as such, serve as natural substrates of the 20S proteasome. Moreover, unlike the 26S proteasome, which consumes ATP for substrate unfolding prior to degradation, the 20S proteasome activity does not depend on ATP, which is depleted during oxidative stress, thus lessening the functionality of the 26S proteasome [14,15]. Furthermore, the ubiquitinylating and deubiquitinase enzymes are reversibly inactivated upon oxidation by ROS, which, together, reduce the efficacy of the 26S proteasome degradation route [31,32,33].

After oxidative insult, HSP70 not only acts as a 19S holdase, but it was also shown to interact with both the 20S proteasome and oxidized protein substrates. These interactions were found to increase the 20S-mediated degradation of oxidized proteins, possibly by shuttling the substrates toward the 20S proteasome for degradation [34]. Moreover, redox regulation is also reflected in the multiple oxidation-driven modifications that modulate 20S proteasome activity. Among them are changes in proteasome subunits via the lipid peroxidation derivative 4-Hydroxynonenal (4-HNE), carbonylations, *S*-glutathionylations, and glycoxidations [35]. In several cases, oxidation-related post-translational modifications were shown to directly affect 20S proteasome activity. In K562 cells, for example, exposure to hydrogen peroxide caused modification of nuclear 20S proteasomes by poly-ADP ribosylation, which was accompanied by an increase in the proteolytic capacity of the 20S proteasome toward oxidatively damaged histones [36]. In yeast, *S*-glutathionylation of cysteines on the α5 subunit of 20S proteasomes was shown to induce an opening of the 20S proteasome gate and increase the degradation capacity of the complex [25].

Further support for the involvement of the 20S proteasome in combating oxidative stress comes from a study in plants. It was shown that several *Arabidopsis thaliana* lines of weak loss-of-function mutations in 19S proteasome subunits exhibited higher levels of free 20S proteasomes, and an increase in the catalytic capacity of the complex. The rise in 20S proteasome levels in the mutant lines, compared to wild type (WT), enhanced the cellular ability to degrade oxidized proteins and, thus, increased the oxidative stress tolerance [37]. Overall, it is likely that 26S and 20S proteasome activities are needed for different cellular states, and that under oxidative stress, higher levels of the 20S proteasome are beneficial.

In addition to the direct, short-term effects of oxidative stress on the levels and activity of the 20S proteasome, cells activate an adaptive homeostasis mechanism that results in transient changes in gene expression and increased stress resistance [38]. It was found that prolonged adaptation of cells to elevated levels of oxidizing agents over 12–24 hours induces significant upregulation in the expression of 20S proteasome subunits, whereas the levels of 19S regulatory particle subunits are not affected [23]. This upregulation is primarily mediated by Nrf2, the master regulator of oxidative stress [39,40,41]. In addition to the upregulation of the 20S proteasome subunits, an increase in immunoproteasome expression, as well as the alternative 20S proteasome regulators PA28αβ PA28γ and PA200 have also been observed during inflammatory processes and oxidative stress [40,42]. Association between the 20S proteasome and the PA28αβ and PA28γ regulators enhanced the degradation of oxidized proteins. PA28αβ also increased the ability of the immunoproteasome to degrade oxidized proteins. Interaction of the complex with PA200, on the other hand, reduced the proteolytic capacity of both the housekeeping and the immuno-20S proteasomes [42], suggesting that the interplay between the different proteasomes and their regulatory complexes results in a coordinated mechanism that controls the degradation of oxidized proteins.

During aging, particularly through the last third of the organism’s lifespan, the capacity for adaptive homeostasis gradually declines. This, in turn, reduces the organism’s ability to resist oxidative stress insults [38]. The decline in adaptive homeostasis is reflected in both a gradual reduction in the organism’s ability to upregulate proteasome subunits, and a decrease in the proteolytic capacity of the 20S proteasome in response to oxidative stress. This deterioration in 20S functionality impacts adaptation to oxidative stress in aged cells [43,44]. Similarly, different neurodegenerative diseases such as Alzheimer’s and Parkinson’s that are linked to the aging process, were also shown to be affected by dysfunctional 20S proteasomes [45,46]. However, the exact role of the 20S particle in these diseases remains unclear.

Recently, an additional means of regulating the 20S proteasome has been discovered. The two Rossman fold-containing proteins NQO1 and DJ-1 were shown to specifically bind the 20S proteasome, though not the 26S proteasome, and inhibit its activity, rescuing 20S proteasome substrates from degradation [39,47,48]. Under conditions of oxidative stress, NQO1 and DJ-1, together with Nrf2 and the 20S proteasome, form a regulatory circuit. Within this feedback loop, DJ-1 acts as a bifunctional protein, promoting not only the inhibition of 20S proteasome, but also its activation. DJ-1 prevents protein degradation by physically binding to the 20S proteasome, as well as by inducing the expression of NQO1 through Nrf2 stabilization. NQO1, like DJ-1, directly binds the 20S proteasome, lowering its activity. In parallel, Nrf2-mediated expression of the 20S proteasome occurs. This mechanism enables control over the activity of the 20S proteasome, which keeps the balance between the need to rapidly degrade oxidized proteins in response to stress, while, on the other hand, prevents excessive degradation of intrinsically unstructured proteins that are important factors in major regulatory and signaling events.

## 3. Involvement in Neuronal Communication

While it is generally accepted that 20S proteasomes are localized to the cytoplasm and the nucleus, a recent landmark study discovered the presence of 20S proteasomes within the plasma membrane of neurons [49]. These embedded proteasomes, termed nuclear membrane proteasomes (NMPs), degrade intracellular proteins, producing extracellular peptides that stimulate neuronal activity. A deeper analysis of the peptides produced by NMPs indicated that they include nascent ribosome-associated polypeptides of immediate early genes (IEGs) such as c-Fos and Npas4, which are proteins that are characterized by rapid gene transcription upon neuronal stimulation, and are critical to the development and functioning of the nervous system [50]. This finding demonstrates a close association between the NMPs and ribosomes, enabling co-translational degradation of the nascent polypeptides during such stimulation. Notably, this process was shown to be independent of ubiquitination, and occurred in the absence of any identified proteasome regulatory caps, further confirming that this neuronal signaling function is a 20S proteasome-specific phenomenon. Many questions remain to be addressed in this exciting new field, such as the mechanism by which the NMP is anchored to the neuronal membrane, how the ribosome and IEG mRNA are specifically targeted to the NMPs to facilitate co-translational degradation, and whether 20S proteasomes are embedded within other cell membrane types, and take part in cell-cell communication.

## 4. Post-Translational Processing

The outcome of proteolysis by the 20S proteasome is not restricted to the production of peptides since proteins are not always entirely degraded. Several studies have demonstrated that cleavage products are generated from proteins that are either partially degraded, or endo-proteolytically processed by the 20S proteasome. These cleaved protein products display altered functionality compared with that of their parent proteins, affecting their associated cellular processes and/or interaction partners [51]. One of the first reported cases of 20S proteasomal processing involved the translation initiation factors eIF3 and eIF4F, which bind to the 40S ribosomal subunit or mRNA, respectively. This initiates the recruitment of the ribosome to the mRNA. Subunits of these protein complexes, eIF3a and eIF4G, were shown to be specifically cleaved by the 20S proteasome at defined and presumably disordered sites, yielding distinct protein fragments, while other subunits remained intact [52]. This proteolysis affected the functional assembly of the ribosomal pre-initiation complexes, inhibiting the translation of certain mRNAs.

Various transcription factors have also been shown to be selectively cleaved by the 20S proteasome. Y-box protein 1 (YB-1) is known to interact with both mRNA in the cytoplasm and DNA in the nucleus, shuttling between these two cellular compartments as required. YB-1 contains both nuclear and cytoplasmic localization signals in its C-terminal domain, and selective cleavage by the 20S proteasome of the final 105 amino acids (which contain the cytoplasmic localization signal and are intrinsically disordered) facilitates its transition to the nucleus [53]. This limited proteolysis was triggered by drug-induced genotoxic stress, which leads to the accumulation of truncated YB-1 in the nucleus, where it can activate the transcription of DNA repair genes. In addition, p50, a member of the NF-κB transcription factor family, is produced by endoproteolytic processing of its precursor, p105, by the 20S proteasome [54]. While p105 forms a dimer, the C-terminus is relatively unstructured and remains exposed, enabling 20S proteasome-mediated proteolysis. Removal of the C-terminus occurs only up to a glycine-rich region, which protects the rest of the protein and results in the production of p50. The discovery of this processing mechanism overturned the previously-held theory that p50 is produced by ribosomal pausing by means of co-translation.

p53, the so-called “guardian of the genome”, is a transcription factor known to exist in multiple isoforms, each produced by varying mechanisms, including alternative promotor usage and splicing. One of these isoforms, Δ40p53, which lacks the first 39 residues of full-length p53, can be produced in one of two ways. The first is via alternative translation initiation, and the second, recently discovered mechanism, is via 20S proteasomal processing [55]. The p53 N-terminus is intrinsically disordered, and can, therefore, be cleaved by the 20S proteasome, which generates the Δ40p53 isoform. This isoform can be incorporated into a hetero-tetramer with full-length p53, reducing p53’s transcriptional activity. This negative regulation of p53 activity is particularly pronounced under oxidative stress, due to an increase in Δ40p53 generation as a result of increased 20S proteasome activity [55].

The ubiquitin-like protein LC3 (microtubule-associated protein 1B Light Chain-3), a key player in the autophagic pathway, has also been shown to be processed by the 20S proteasome [56]. LC3 binds to phosphatidylethanolamine (PE) to create a conjugate that is later recruited to the autophagosomal membranes [57]. In vitro processing of LC3 by the 20S proteasome led to cleavage between residues 66–77, generating a product termed LC3T that lacks the ubiquitin fold and is no longer capable of binding PE. While no function has yet been assigned to LC3T, this processing is negatively regulated by p62, a LC3-binding protein participating in autophagic degradation of polyubiquitinated protein aggregates [56]. These observations suggest a possible link between the 20S itself and autophagy, warranting further research to fully understand the significance of this connection.

The central chaperone HSP70, which is particularly active in controlling protein quality following environmental stresses, can also be endo-proteolytically cleaved by the 20S proteasome, which removes the unstructured C-terminus, and produces a stable 30 kDa fragment from the N-terminus of the protein [58]. The functional significance of this processed fragment has yet to be determined. Considering HSP70’s numerous essential roles in protecting cells from damage, coupled with its role in 20S proteasome activation following oxidative insult, clarifying the potential influence of this fragment is critical to the understanding of HSP70 biology.

One feature common to all of these processing events is that they occur at disordered regions within the protein’s structure, and, in most cases, end at the boundary between disorder and order. Thus, cleavage of these IDRs by the 20S proteasome reinforces our understanding of the substrate selectivity of the 20S proteasome. The ability of the 20S proteasome to specifically cleave fragments of its substrate proteins to modulate protein function and/or alter downstream processes constitutes an exciting phenomenon that may have broad implications for cell function and survival. Future research is expected to shed light on the regulatory principles of this process, and the repertoire of processed substrates.

## 5. The Extracellular 20S Proteasome

Traditionally, proteasomes were mainly studied as intracellular proteases. However, in recent years, a growing body of evidence indicates that active 20S proteasomes are also located in extracellular fluids [59]. Not much is known about the nature of these proteasomes, their origins, compositions, and substrate specificities. Here, we will examine the current knowledge of two types of extracellular 20S proteasomes: those circulating freely in the plasma (c-20S), and those encapsulated in extracellular vesicles (EV-20S).

### 5.1. Circulating 20S Proteasomes

Circulating 20S proteasomes (c-20S) are physiologically present in human plasma, and their levels increase significantly under a variety of pathological conditions, including autoimmune diseases, cancerous tumors, trauma, sepsis, and acute respiratory distress [60,61,62,63,64,65]. Electron microscopy studies determined that the complexes are intact [66]. Moreover, proteomic analyses indicated that the complexes consist exclusively of 20S subunits [67,68], lacking components of the 19S regulatory particle [67,68]. While the precise physiological role of c-20S is unclear, their elevated levels were shown to correlate with the progression of blood cancers, solid tumors [69,70], and autoimmune conditions [71]. Moreover, high levels of c-20S were measured after burn injury. The highest levels were detected at day zero, with a gradual decline over the course of a week [72]. Therefore, levels of 20S proteasomes have been proposed to serve as prognostic biomarkers for both disease states and/or treatment efficacy [59].

The origins of c-20S remain unclear. However, it is thought that they are secreted from the cells themselves by an as-yet unidentified mechanism [73,74]. Efforts were also undertaken to define c-20S proteasome function and substrates. The c-20S proteasome present in the alveolar space was capable of degrading albumin, possibly assisting in the maintenance of low intra-alveolar oncotic pressure [75]. Another recent study identified osteopontin (OPN) as a substrate of the c-20S proteasome in multiple sclerosis [74]. OPN is an extracellular matrix protein, which plays a role in numerous pathologies such as inflammation, cancer, diabetes, renal stones, and cardiovascular disease [76]. OPN was found to be processed by the c-20S proteasome, yielding biologically active peptides. An in vitro degradation assay confirmed that OPN processing is 20S proteasome-dependent, and that the resulting fragments possess chemotactic activity [76]. In particular, a feedback loop is generated between the OPN peptides and c-20S proteasomes, whereby the c-20S generated peptides prevent further proteasome release into the extracellular space by endothelial cells. These findings reinforce the hypothesis that molecular and cellular cues function in tandem to coordinate c-20S secretion.

### 5.2. 20S Proteasome within Extracellular Vesicles

Extracellular vesicles (EVs), which are 30–500 nm lipid bilayer vesicles secreted by almost every cell type, have been recognized as vehicles of intercellular communication under both physiological and pathological conditions [77,78]. Their cargos include proteins, lipids, and RNA that are selectively recruited and packed within, before being released into the extracellular environment. EVs are increasingly being recognized as contributing factors in many diseases, and their potential as diagnostic and prognostic markers, as well as therapeutic targets, is beginning to emerge.

A search in the ExoCarta database (http://www.exocarta.org/), a manually curated database of exosomal proteins, RNA, and lipids, indicated the presence of 20S proteasome subunits in EVs isolated from a variety of cell lines. Furthermore, detailed proteomic analysis identified all the subunits comprising the 20S proteasome in EVs isolated from mouse models of prostate cancer [79] and from T lymphocytes grown in culture medium [80]. High levels of active 20S proteasomes were also identified in EVs isolated from tumor-associated macrophages but not from naïve macrophages, which indicates that such high levels of this proteasome might have a specific biological role in the EVs or their target cells. Overall, considering the EV’s restricted volume, it is plausible that 20S-mediated degradation is favored over 26S proteasome ubiquitin-dependent proteolysis, both due to the smaller size of the 20S proteasome, and its independence of ubiquitinylating enzymes.

A recent study linked the activity of EV-20S proteasomes isolated from apoptotic cells with induction of autoantibody production, known to accelerate organ rejection after transplantation [81]. The proteomic analysis performed indicated significant enrichment of 20S proteasome subunits in exosome-like vesicles, but not in apoptotic bodies. Moreover, proteasomal inhibition reduced the autoimmune response. Overall, this study hints at a direct role for the 20S proteasome within the EVs themselves. However, conflicting evidence also suggests that EVs may serve as a “vehicle” for carrying the 20S proteasome to the target tissues [82]. EVs isolated from mesenchymal stem cells were shown to reduce tissue damage after myocardial injury, upon their internalization by the target tissue [83]. Detailed proteomic analysis of the isolated EVs revealed that all subunits of the 20S proteasome, including the three β-subunits of the immunoproteasome, were present, and that the isolated 20S particles were active. Furthermore, EVs taken up by damaged cells reduced the amount of oligomerized proteins, which suggests a possible role for the EV-20S proteasome in the injured target tissue [82]. Overall, the field of extracellular proteasomes falls into relatively uncharted territory since, until now, the proteasome community has focused primarily on intracellular proteasomes. Many specific questions regarding these 20S proteasome species, such as their origin, composition, regulation, and substrate specificities, have yet to be addressed.

## 6. Degradation of Disordered Proteins

It is estimated that more than 40% of human protein-coding genes contain IDRs, which makes them susceptible to 20S proteasome degradation [84]. This group includes numerous proteins with signaling and regulatory functions. However, identifying the entire repertoire of substrates degraded by the 20S proteasome constitutes a challenging task, since different pools of the same protein can be targeted for degradation via either the 20S or the 26S proteasome [85,86]. A need, therefore, exists to uncouple these two pathways, in order to expose 20S-specific substrates.

In the past, individual case studies have pointed to proteins susceptible to 20S-mediated degradation (reviewed in Reference [5]). For example, the tumor suppressors p53, p73, and retinoblastoma protein, the proto-oncoprotein c-Fos, the cell cycle regulators p27 and p21, and the neurodegenerative disease-related proteins tau and α-synuclein [5,11,87] have been identified as substrates of the 20S proteasome [9]. Interactions of such IDRs with partner proteins can, however, protect them from 20S proteasomal activity [84]. The 20S proteasome substrate landscape is further broadened by the fact that feeding the proteins into the 20S proteasome is not restricted to the proteins’ termini, since evidence exists for endoproteolytic activity of internal disordered regions as well [88].

A recent attempt to systematically define the repertoire of 20S proteasome substrates was undertaken through proteomic analysis of heat-treated HeLa nuclear-rich extracts subjected to degradation by purified 20S proteasomes [89]. Hundreds of such substrates were identified in this study many of which were found to be proteins involved in the formation of phase-separated granules. Although these findings are biased towards heat-soluble nuclear IDPs, they support the view that a substantial proportion of the proteome is susceptible to 20S proteolysis. These results may represent only the leading edge, as many additional 20S substrates await discovery.

## 7. Spliced Peptide Generation

Antigenic peptides displayed by HLA class I molecules are primarily derived from proteasomal degradation of cellular proteins. These peptides were originally believed to be exclusively linear fragments of the degraded proteins. However, the discovery of non-contiguous peptides derived from fragments distant in the primary sequence of the parent protein uncovered a potential new source of antigenic peptides [90,91]. In this process, the proteasome not only cuts proteins into fragments, but it is also thought to ligate them through a process named proteasome-catalyzed peptide splicing, i.e., generating a peptide bond between two non-contiguous fragments of a polypeptide substrate, thereby producing a spliced peptide with a sequence not present in the original protein. The catalytic subunits of the 20S proteasome attack the carbonyl group of a peptide bond, generating an acyl-enzyme intermediate. The acyl-enzyme can then be hydrolyzed by a water molecule, producing the C-terminal end of the peptide, and enabling its release from the catalytic chamber. During peptide splicing, however, the free N-terminal amino group of a second peptide present in the chamber reacts with the acyl-enzyme intermediate instead of a water molecule, which leads to the formation of a new peptide bond in a process known as transpeptidation [91,92].

This process can occur between peptides that are separated by only a few amino acids in the parental protein, such as the melanocytic glycoprotein gp100 derived peptide [91], as well as those that are much more distant; e.g., 40 amino acids apart, as observed for the fibroblast growth factor 5 (FGF-5)-derived peptide [90,92]. Transpeptidation can even result in reverse splicing—the joining of two peptides in the opposite order in which they existed in the parent protein, as for example, is seen in the case of the SP110 nuclear phosphoprotein peptide [93]. In all of these cases, peptide production was observed using purified 20S proteasomes, and occurred independently of ubiquitination.

The formation and presentation of spliced peptides is considered a relatively rare occurrence even though one recent study of the HLA-I immunopeptidome suggested that spliced peptides represent one-third of the diverse population of bound peptides, and comprise roughly 25% of total HLA-I bound peptide abundance [94]. These proposed proportions of 20S proteasome-generated spliced peptides are intriguing. However, further research is needed to fully understand the extent and relevance of peptide splicing to the immune processes.

## 8. Conclusions and Perspectives

Initially, protein degradation by means of the 20S proteasome was questioned, since it was believed that proteins are only eliminated by the 26S ubiquitin-proteasome system. However, emerging evidence, summarized here, emphasizes the significance of the 20S proteasome complex as an independent cellular machine. The notion that arises is that the 20S proteasome is not only a clearance and recycling entity, but also acts as a regulator of cellular processes. In particular, findings indicate that the 20S proteasome complex can alter protein function by means of selective post-translational processing [51], and mediate neuronal communication by co-translational cleavage of nascent proteins [49,50]. The rapid progress over the past decade in uncovering the variety of molecular mechanisms involving the 20S proteasome likely constitutes the tip of the iceberg. We expect that ongoing research will expose more regulators of the proteasome, such as activators and inhibitors that modulate the activities of the complex, and novel downstream targets that are either cleaved or degraded by it (Figure 2). It is also possible that, analogous to the 26S proteasome system, shuttling proteins that capture substrates and escort them to the 20S proteasome may exist. Moreover, the increasing pace of activity in the field augurs well for the discovery, in the not-too-distant future, of additional factors affecting the interactions, abundance, and localization of this protease.

Current knowledge of 20S proteasome substrates suggests that this group of proteins can be divided into two categories: (i) functional and (ii) pathological entities. Among the functional proteins are those containing IDRs. This set of proteins is dominated by those that interact with or function as hubs in protein interaction networks, performing central roles in the regulation of signaling pathways and crucial cellular processes [95]. Pathological proteins comprise those that in their native state are folded, but aging, mutations or oxidative stress have resulted in a partial loss of their structure. This latter set of proteins is prone to aggregation, and may lead to the development of various pathologies like neurodegenerative disorders such as Alzheimer’s, Parkinson’s disease, and Huntington’s disease. To date, it remains unclear whether specialized regulatory mechanisms have evolved to orchestrate 20S proteasome degradation of each class of substrates, or whether intricate networks of regulators exist. Systematic studies encompassing the full 20S proteasome substrate landscape will likely clarify the regulatory architecture underlying the complex’s activity.

As biochemical methods and analytical tools are improving, it is becoming clear that the 20S proteasome is composed of a diverse population of functionally distinct entities (reviewed in References [96,97]). This diversity is generated by the three-active site-containing β-subunits that can all, or in part, be replaced by immuno-subunits, leading to the formation of intermediate or mixed-type proteasomes displaying altered protein cleavage specificities. Their heterogeneity is further increased by the identification of tissue-specific 20S species, including the thymoproteasome and spermatoproteasome, which include specific β_5_ and α_4_ subunits, respectively [98,99]. An additional impressive layer of complexity is presented by diversity within individual subunits due to genetic variations, alternative splicing, and numerous post-translational modifications [96]. Many issues regarding the impact of the modulations resulting in this striking heterogeneity remain relatively unexplored. For example, we do not know whether the various 20S proteasome subtypes associate with the 19S complex to maintain the 26S/20S balance, or if there is a preference for one state or the other. In addition, the different modulations may affect the catalytic activity of the complex, its affinity toward substrates, or its interaction with regulators. Furthermore, the diverse variants of the 20S proteasome may be located in different cellular or extracellular compartments or tissues, and novel tissue-specific α or β subunits are yet to be discovered. We anticipate that expanding our knowledge in these areas may pave the way towards a deeper understanding of the functioning of the complex.

Lastly, as details related to 20S proteasomal activity are clarified, its links to diverse diseases and aging are coming into focus. As discussed above, high levels of extracellular proteasomes, both those circulating in plasma and those in extracellular vesicles, are correlated with a range of pathologies, cancer included, while gradual loss of 20S proteasomal function is associated with aging and neurodegenerative diseases. These observations exemplify the importance of in-depth exploration of the 20S proteasome degradation pathway, to determine its potential therapeutic efficacy. To date, proteasome inhibitors such as bortezomib, carfilzomib, and ixazomib have been developed for treating certain cancers, especially multiple myeloma and mantle cell lymphoma and many other such inhibitors are currently being tested for their anti-tumor and anti-inflammatory effects, as well as for treating autoimmune diseases [100,101]. These drugs, however, target the chymotrypsin-like activity of the 20S proteasome, and inhibit the activities of both the 20S and 26S proteasomes. Thus, by utilizing the accumulating knowledge on selective 20S proteasome inhibitors, the smart design of selective drug interventions specifically inhibiting the 20S proteasomes over 26S proteasomes will improve the rates of cancer cell toxicity, and/or minimize the deleterious side effects of the current therapeutic regimens and expand their applications. Furthermore, 20S proteasomal activation is expected to prove its relevance in the clinic, in instances of disease caused by toxic or mutated proteins common in aging, as well as in neurodegenerative ailments and cardiomyopathies.

## Figures and Tables

**Figure 1 biomolecules-09-00190-f001:**
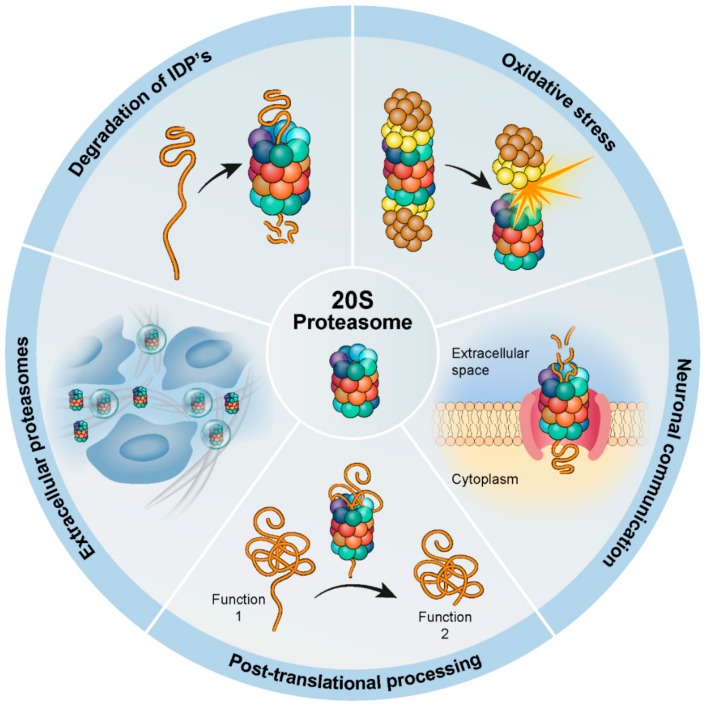
A scheme summarizing the various biological systems and pathways influenced by 20S proteasome activity. Multiple cellular processes are influenced by the 20S proteasome, such as the degradation of intrinsically disordered proteins (IDPs), the oxidative stress response during which the 19S regulatory particle dissociates from the 20S proteasome while leaving 20S proteasome activity intact, and the production of peptides at the neuronal membrane to facilitate neuronal communication. In addition, the 20S proteasome post-translationally processes certain proteins, leading to products with altered functionality compared with the parent protein. Furthermore, the discovery of 20S proteasomes in the extracellular space, both free circulating 20S proteasomes and those found within extracellular vesicles, indicate an involvement in a broad range of biological pathways beyond the confines of the cell.

**Figure 2 biomolecules-09-00190-f002:**
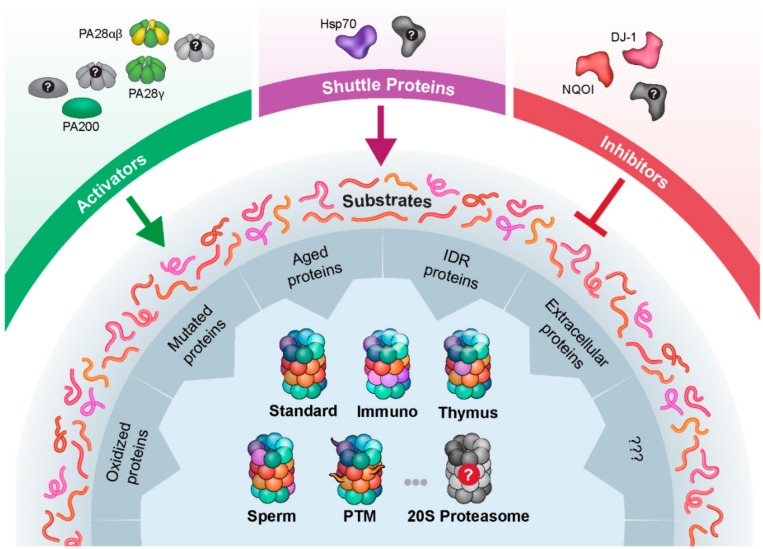
There are many unknown elements in the 20S proteasome degradation pathway. The scheme summarizes the different substrate groups, 20S proteasome forms, activators, inhibitors, and shuttle proteins, and point toward the many entities among these players that are yet to be discovered. HSP70: heat shock protein 70, IDR: intrinsically disordered proteins, PTM: post translational modification.

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
