# Peer review of "The Contribution of the 20S Proteasome to Proteostasis"

_biomolecules, 2019, doi:10.3390/biom9050190_

Reviewer 1 Report

The authors provide a useful and timely reflection on the knowledge of possible 19S RP-independent functions of the 20S proteasome core particle (CP). 

The notion that CPs are capable of degrading proteins with unstructured domains is largely based upon in vitro studies. As the authors rightly point out, it is an at the same time difficult to address and important question whether CPs perform a similar 19S RP-independent function in vivo. Therefore, the scarce and partly conflicting evidence addressing e.g. the role of the 19S RP in the degradation of oxidatively damaged proteins and the resistance to oxidative stress deserves special attention. Genetic evidence from plants indicated that mutations in 19S RP components augment resistance to oxidative stress (Kurepa, Toh-e, and Smalle, The Plant Journal 2008, 53, 102-114; not cited in this manuscript). Even though this could also be explained by other cellular responses, this observation is at least consistent with the idea that 20S CPs are well suited to act in vivo on oxidatively damaged proteins in an 19S RP-independent manner. Similarly, it was observed that deletion of the ECM29 gene in yeast interfered with the dissociation of CP and RP upon oxidative stress, and, at the same time, reduced resistance to hydrogen peroxide (reference 26). Contrasting these findings, it was observed that mouse cells with a KIAA0368 ablation showed increased resistance to hydrogen peroxide along with reduced dissociation of CP and RP, suggesting that 26S proteasomes are capable of dealing with the damaged proteins that occur under these conditions (reference 27). While these findings leave questions open, I find them worth reflecting upon in more detail in the context addressed in this manuscript.

I find it also worthwhile to consider mentioning that Cdc48 may act as an alternative ATPase complex cooperating with the CP (e.g. Barthelme and Sauer, PNAS 2013, 110, 3327-3332).

Overall, this is a well-written and balanced review.

Minor point:

There is a typo in line 249: “unde-rtaken”.

Author Response

Reviewer 1

1. The notion that CPs are capable of degrading proteins with unstructured domains is largely based upon in vitro studies. As the authors rightly point out, it is an at the same time difficult to address and important question whether CPs perform a similar 19S RP-independent function in vivo. Therefore, the scarce and partly conflicting evidence addressing e.g. the role of the 19S RP in the degradation of oxidatively damaged proteins and the resistance to oxidative stress deserves special attention. Genetic evidence from plants indicated that mutations in 19S RP components augment resistance to oxidative stress (Kurepa, Toh-e, and Smalle, The Plant Journal 2008, 53, 102-114; not cited in this manuscript). Even though this could also be explained by other cellular responses, this observation is at least consistent with the idea that 20S CPs are well suited to act in vivo on oxidatively damaged proteins in an 19S RP-independent manner. Similarly, it was observed that deletion of the ECM29 gene in yeast interfered with the dissociation of CP and RP upon oxidative stress, and, at the same time, reduced resistance to hydrogen peroxide (reference 26). Contrasting these findings, it was observed that mouse cells with a KIAA0368 ablation showed increased resistance to hydrogen peroxide along with reduced dissociation of CP and RP, suggesting that 26S proteasomes are capable of dealing with the damaged proteins that occur under these conditions (reference 27). While these findings leave questions open, I find them worth reflecting upon in more detail in the context addressed in this manuscript?

We thank the reviewer for highlighting this point and directing us to the relevant references. In the revised version, we added a paragraph emphasizing the study performed in Arabidopsis thaliana, in which lines of weak loss-of-function mutants of 19S proteasome subunits, exhibited higher levels of free 20S proteasomes with increased tolerance against oxidative stress. We also elaborate on the functions of Ecm29/KIAA0368 and emphasize the remaining open question regarding the functionality of the 26S proteasome under oxidative stress.

2. I find it also worthwhile to consider mentioning that Cdc48 may act as an alternative ATPase complex cooperating with the CP (e.g. Barthelme and Sauer, PNAS 2013, 110, 3327-3332).

We appreciate the reviewers suggestion and have indeed read with great interest the Barthelme and Sauer, PNAS 2013 manuscript. However, after thoughtful consideration we decided not to include this study in the manuscript, as emphasis was not given here to any of the alternative 20S proteasome activators that deserve an independent and detailed review on their own.

3. There is a typo in line 249: “unde-rtaken”.

We thank the review for spotting this typo that has been corrected accordingly.

Reviewer 2 Report

26S proteasome has been extensively studied as it plays central role in maintaining cellular homeostasis in physiological and pathological processes. In this review, the authors focused on the role of 20S proteasome, which is a component of the canonical 26S proteasome. It is a very good angle to approaching the contribution of 20S proteasome and summarizing the up-to-date studies on this subject. The manuscript is well written and properly organized. The function of 20S proteasome is thoroughly reviewed and the future directions are fully discussed. Overall it is a good review, which I think will contribute to the understanding of the delicate network of protein degradation.

There are a few thoughts and suggestions which I would like to share in the hope of improving the integrity and readability of the manuscript.

1.     Figure1: the content of Figure 1 is discussed throughout the text, but it would be nice if authors can add figure legend to give readers a chance of quick view of the entire idea.

2.     What is the difference between “spliced peptide generation” and “post-translational processing? Is the former one type of the latter?

3.     Is there any study on the crosstalk of 20S proteasome and autophagy?

4.     Line 249: typo “undefallrtaken”.

5.     The selective drug specifically inhibiting 20S is discussed, which I think could be challenging. As 20S is the catalytically active component of 26S proteasome, a small molecule which inhibits 20S would be likely to inhibit 26S as well. I would like to see a smart design to distinguish these two in the future.

Author Response

Reviewer 2

26S proteasome has been extensively studied as it plays central role in maintaining cellular homeostasis in physiological and pathological processes. In this review, the authors focused on the role of 20S proteasome, which is a component of the canonical 26S proteasome. It is a very good angle to approaching the contribution of 20S proteasome and summarizing the up-to-date studies on this subject. The manuscript is well written and properly organized. The function of 20S proteasome is thoroughly reviewed and the future directions are fully discussed. Overall it is a good review, which I think will contribute to the understanding of the delicate network of protein degradation.

There are a few thoughts and suggestions which I would like to share in the hope of improving the integrity and readability of the manuscript.

1.     Figure1: the content of Figure 1 is discussed throughout the text, but it would be nice if authors can add figure legend to give readers a chance of quick view of the entire idea.

A figure legend explaining the schematic of Figure 1 has been added to aid the readers.

2.     What is the difference between “spliced peptide generation” and “post-translational processing? Is the former one type of the latter?

Spliced peptide generation specifically refers to the process of transpeptidation that occurs within the 20S proteasome in which non-contiguous fragments of a polypeptide substrate or a protein are ligated, thereby generating a spliced peptide with a sequence not present in the original protein. This presumably occurs during degradation of the parent protein, yielding original peptides for immune display. Post-translational processing however refers to partial cleavage of certain proteins, yielding a cleaved but active product with altered functionality. The text has been revised accordingly to clarify the difference between the two processes.

3.     Is there any study on the crosstalk of 20S proteasome and autophagy?

We thank the reviewer for pointing this out. We have found only one study that demonstrated a possible connection between the 20S proteasome and autophagy. Gao et al emphasize the ability of the 20S proteasome to partially cleave LC3, a ubiquitin like protein, key in the autophagy process. In the revised manuscript, we have included this work in the “Post-translational Processing” section, specifying the possible crosstalk link between autophagy and 20S proteasome degradation.

4.     Line 249: typo “undefallrtaken”.

We thank the review for spotting this typo that has been corrected accordingly.

5.     The selective drug specifically inhibiting 20S is discussed, which I think could be challenging. As 20S is the catalytically active component of 26S proteasome, a small molecule which inhibits 20S would be likely to inhibit 26S as well. I would like to see a smart design to distinguish these two in the future.

We agree with the reviewer, it is indeed a challenge to specifically inhibit the 20S proteasome over the 26S, and the careful design of inhibitors is especially important. We believe that proteins like DJ-1 and NQO1 that specifically inhibit the 20S proteasome, but not the 26S counterpart, may provide the molecular clues for selective 20S inhibition. To further emphasize this we have revised the relevant paragraph accordingly.